# Single-Cell Analysis of Circulating Tumor Cells: Why Heterogeneity Matters

**DOI:** 10.3390/cancers11101595

**Published:** 2019-10-19

**Authors:** Su Bin Lim, Chwee Teck Lim, Wan-Teck Lim

**Affiliations:** 1NUS Graduate School for Integrative Sciences & Engineering, National University of Singapore, Singapore 117456, Singapore; sblim@u.nus.edu (S.B.L.); ctlim@nus.edu.sg (C.T.L.); 2Department of Biomedical Engineering, National University of Singapore, Singapore 117583, Singapore; 3Mechanobiology Institute, National University of Singapore, Singapore 117411, Singapore; 4Institute for Health Innovation and Technology (iHealthtech), National University of Singapore, Singapore 117599, Singapore; 5Division of Medical Oncology, National Cancer Centre Singapore, Singapore 169610, Singapore; 6Office of Academic and Clinical Development, Duke-NUS Medical School, Singapore 169857, Singapore; 7IMCB NCC MPI Singapore Oncogenome Laboratory, Institute of Molecular and Cell Biology (IMCB), Agency for Science, Technology and Research (A*STAR), Singapore 138673, Singapore

**Keywords:** single-cell analysis, cellular heterogeneity, circulating tumor cells

## Abstract

Unlike bulk-cell analysis, single-cell approaches have the advantage of assessing cellular heterogeneity that governs key aspects of tumor biology. Yet, their applications to circulating tumor cells (CTCs) are relatively limited, due mainly to the technical challenges resulting from extreme rarity of CTCs. Nevertheless, recent advances in microfluidics and immunoaffinity enrichment technologies along with sequencing platforms have fueled studies aiming to enrich, isolate, and sequence whole genomes of CTCs with high fidelity across various malignancies. Here, we review recent single-cell CTC (scCTC) sequencing efforts, and the integrated workflows, that have successfully characterized patient-derived CTCs. We examine how these studies uncover DNA alterations occurring at multiple molecular levels ranging from point mutations to chromosomal rearrangements from a single CTC, and discuss their cellular heterogeneity and clinical consequences. Finally, we highlight emerging strategies to address key challenges currently limiting the translation of these findings to clinical practice.

## 1. Introduction

The concept of intratumoral heterogeneity (ITH), first described in 1982 by Fidler and Hart [1], has been expanded to include genetic, phenotypic and functional heterogeneity within tumors comprising diverse malignant and non-malignant subpopulations. With accumulations of mutations in DNA damage checkpoint control genes and DNA repair genes, divergent cancer clones may evolve and propagate over time through selection processes driven by constantly changing microenvironment and by the use of therapy [2,3]. In addition to genetic heterogeneity in clonal mutations and subclonal de novo mutations, functional heterogeneity related to developmental pathways and epigenetic programs and spatial variability in tumor microenvironment contribute to ITH, which governs key aspects of tumor biology, including tumor invasion, metastasis, and drug resistance [2].

Recent years have also seen the contribution of non-malignant cells, such as stromal fibroblasts, immune cells, bone-marrow-derived cells, mesenchymal stem cells, and endothelial cells, to ITH within a tumor. Malignant and benign cells interact locally through a complex network of extracellular matrix (ECM) and the related components, collectively termed as matrisome, which has been linked to tumor progression, response to adjuvant therapy, and immune system [4,5,6,7], driving tumor phenotypes supporting their metastatic competence. The tumor cells can be shed passively (through corrupted blood vessel) and/or actively (through epithelial–mesenchymal transition (EMT)) from tumors into the circulation, referred to as circulating tumor cells (CTCs), in which only a minority of cell populations survive under the physiological blood flow and succeed in early colonization phases [8]. Despite the low frequency of occurrence, CTCs allow for repeated sampling, which is not clinically practical for tissue biopsy, and may thus be an excellent tool for assessing tumor heterogeneity and for revealing clonal diversity underlying resistance to treatment [9].

The prevalence of phenotypic plasticity involving programmed death-ligand 1 (PD-L1) [10], stemness [11], drug resistance [12], and EMT [13], within CTC populations has fueled investigation of cellular heterogeneity using single-cell high-throughput enrichment and sequencing platforms over the past five years. Emerging single-cell RNA sequencing (scRNA-seq) data further suggest that CTCs may interact with hematopoietic cells or platelets in blood through direct and/or indirect routes, adding another layer of heterogeneity [14]. Bulk-cell approaches traditionally deployed in existing CTC literature, however, have provided insights into cellular processes averaged throughout the enriched blood sample, which largely comprises leukocytes, or white blood cells (WBCs). The leukocyte contamination is inevitable in any given primarily enriched sample due in part to extremely rare CTCs occurring at a frequency of ~1 in 10^7^ WBCs in blood from a cancer patient [15] and the relatively low cell capture efficiency of existing cell sorting technologies, which are further limited to isolating only certain CTC subpopulations (refer to the Section 2.1 for further details). It is therefore essential to reach single-cell resolution to precisely characterize CTCs at the genomic level and further to investigate the clinical impact of cellular heterogeneity present within CTC populations.

## 2. Methods and Technologies

Despite technical challenges, single-cell CTC (scCTC) analyses have so far revealed genomic variations specific to each CTC, including mutations that are not yet present in the Catalogue of Somatic Mutations in Cancer (COSMIC) database [16,17,18,19,20] or subclonal alterations that are not easily discernible from tissue biopsies [16,17], cells of origin of cancer (e.g., bone marrow-derived multiple myeloma) [21], or with bulk-cell approaches [22], providing comprehensive landscape of evolving tumor cells. Such private genomic variations shared by CTCs may represent “CTC phenotypes”, including intravasation competency, increased migration/motility, enhanced cell–cell interactions, variation in energy metabolism, interaction with platelet and blood immune cells, resistance to anoikis, and resistance to therapy [19,23]. Single nucleotide variations (SNVs) and/or copy number variations (CNVs) in these putative precursors of metastasis present early in clonal evolution or in tumor progression may be excellent targets for therapeutic intervention. Examined below are scCTC DNA sequencing studies that have successfully assessed DNA alterations in patient-derived CTCs across various cancer types (Table 1).

### 2.1. CTC Enrichment

Once collected, blood samples are subjected to CTC enrichment using density gradient centrifugation, 2D/3D microfiltration, microfluidic devices, or immunoaffinity-based technologies (Figure 1; refer to Ref. [14] for detailed summary and comparison of existing CTC enrichment technologies). While CellSearch^®^ remains the choice of primary enrichment tool in scCTC sequencing studies, such immunoaffinity-based enrichment technology relying on epithelial cell surface markers (e.g., EpCAM or CKs) have varying capture efficiency depending on the degree of EMT, stemness, and the resulting differentiation cell state. EpCAM^+^CK^+^ cells have also been detected by CellSearch^®^ in patients with benign diseases, but in lower frequency compared to the cancer group [54]. Clinical data supporting the metastatic competence of EpCAM^−^ CTCs in numerous studies [34,55] have further fueled the transition in the field to the development of label-free approaches leveraging on biophysical properties of CTCs (e.g., size, density, stiffness).

One of our pioneering efforts in developing such marker-independent technology was the application of inertial microfluidics using spiral microchannels, in which depth and width of each channel can be designed to control the positioning of cells or microparticles in suspension via Dean flow fractionation (DFF). Our technology, so-called ClearCell^®^ FX, enriches intact and viable CTCs from the peripheral blood of cancer patients in a fully automated and high-throughput fashion, with a reported >80% sensitivity and specificity in detecting CTCs from clinical samples (refer to [56] for further details on the system). Mechanobiologically inspired enrichment platforms achieve high sensitivity and are not limited to certain CTC subpopulations, yet the purity of enriched samples may be compromised due to their size distribution overlapping with leukocytes, as observed in breast, colorectal, and prostate cancer [57]. To best interrogate CTCs and their cellular heterogeneity, cell enrichment technology should thus be carefully selected for unbiased capture and recovery of different CTC subpopulations.

### 2.2. Single-Cell Isolation

CTC enrichment technologies are used in conjunction with microscopic micromanipulators (e.g., Eppendorf Transfer Man NK2/4 micromanipulator, CellCelector^TM^), laser capture microdissection (LCM) [26], microfluidic devices [44], or DEPArray^TM^ to recover putative CTCs as single cells. Enriched cells are often fixed and stained with the nucleic acid dye DAPI and monoclonal antibodies specific to epithelial cell surface marker cytokeratin (CK) and leukocyte marker CD45, and manually selected by the trained and skilled operator based on DAPI and CK positivity and CD45 negativity. Morphology of the captured cells (e.g., cell shape, size) are concurrently assessed to identify cells having 4 to 40 µm diameter, round or oval shape, and/or high nuclear/cytoplasm ratio [37,39,49].

Even though dozens of CTCs might have been captured in CTC enrichment step, only very few single CTCs (<10 CTCs) have been isolated and transferred successfully to a PCR tube per sample [4,19,49]. Such low recovery rate could be attributed to apoptotic CTCs, which are often excluded from the isolation in most scCTC sequencing studies. Consistent with the classic definition of apoptosis, these cells are defined as CK^+^CD45^-^ CTCs with non-intact nuclei having DAPI pattern of chromosomal condensation and/or nuclear fragmentation and blebbing [52]. Given that low burden of apoptotic CTCs has been associated with poor prognosis and aggressive phenotypes across several cancer types [52,58], characterization of CTC apoptosis in situ may facilitate the development of new platform for real-time monitoring of antitumor drug efficacy.

### 2.3. Whole-Genome Amplification (WGA)

WGA is an active area of development with wide applications to study rare tumor cells or single-celled organisms, such as bacteria and archaea. There exist different WGA approaches based on specific, degenerate, and/or hybrid primers: Linker-adapter PCR (LA-PCR), interspersed repetitive sequence PCR (IRS-PCR), primer extension preamplification (PEP-PCR), degenerate oligonucleotide primed PCR (DOP-PCR), displacement degenerate oligonucleotide primed PCR (D-DOP-PCR), multiple displacement amplification (MDA), single primer isothermal amplification (SPIA), and multiple annealing and looping-based amplification cycles (MALBAC; refer to ref. [59] for a comprehensive summary on working principles and characteristics of each WGA method).

Briefly, LA-PCR and IRS-PCR methods utilize specific primers, each amplifying digested DNA litigated to adapter fragments and repeating sequence elements, respectively, while the rest of the methods, such as PEP-PCR, DOP-PCR, and D-DOP-PCR, are based on the use of degenerated primers. MDA has been commonly applied to single-cell sequencing of microorganisms as well as patient-derived CTCs. It employs a unique polymerase with strong strand displacement activity (e.g., phi29 DNA polymerase), which can amplify fragments of up to 100 kb with high replication fidelity compared to purely PCR-based (e.g., Taq polymerase) methods [59]. MALBAC has been proposed as a hybrid PCR/MDA method, relying on two relatively error-prone DNA polymerases, *Bst* DNA polymerase and Taq DNA polymerase, for isothermal strand displacement and PCR, respectively. Each WGA technique has its own advantages and limitations in terms of sensitivity, specificity, uniformity, and amplification bias. For example, while LA-PCR, DOP-PCR, and MALBAC may be the choice of method for detection of CNVs but not SNVs, MDA (REPLI-g^TM^) has proven to be most sensitive in detecting mutations at a single-base resolution compared to LA-PCR methods (GenomePlex^TM^, Ampli1^TM^) [60].

The challenge is that the yield of amplified DNA varies significantly across CTCs, where the success rate of amplification ranges from 11% to 100% [24,61], and WGA step itself is subjected to coverage biases and errors, such as preferential allelic amplification, GC bias, dropout events, and nucleotide copy errors [60]. To account for such variability, studies have established an additional QC step prior to in-depth sequencing to probe only CTCs with yields of DNA greater than negative controls [24] or a fixed concentration level [27] or those showing specific bands corresponding to targets of interest on the Agilent 2100 Bioanalyzer [19,29]. The author-defined QC assays have also been developed to identify CTCs suited for single-cell targeted sequencing and analysis. For example, genome integrity index (GII), which is determined from detectable PCR bands corresponding to three Mse fragments and KRAS fragment, has been proven to be predictive of successful analysis of sequence-based molecular changes, including point mutations, gene amplifications, and CNVs [30,36,42].

### 2.4. Sequencing and Profiling

Amplified DNA samples are subjected to library preparation and quantification. To date, scCTC studies have most commonly employed next-generation sequencing (NGS), Sanger sequencing, single nucleotide polymorphism (SNP), and array comparative genomic hybridization (aCGH) platforms, and conventional PCR technologies to analyze somatic SNVs, structural variations, (SVs), CNVs, and chromosomal breakpoints and rearrangements for whole exome/genome or selected cancer-associated genes, often comparatively with matched primary tumors and/or metastatic tissues or disseminated tumor cells (DTCs).

In the library QC step, the sequencing depth, percentage of area covered, homogeneity of coverage, and/or SNP densities are assessed to only select high-quality CTC libraries based on author-defined assessment techniques, such as autocorrelation analysis [24] and Lorenz curves [26]. Fluorimetric assays (e.g., Fluorometer) and analytical tool provided by the sequencing platform (e.g., Torrent Suite) may also be used to quantify DNA samples and to assess the performance of sequencing runs and the quality of generated data, respectively [19,31,37]. In some cases, the variants identified by NGS were specifically selected and further validated by Sanger sequencing [31,45] or digital droplet PCR (ddPCR) [36] using the same samples.

The sequence queried in single CTCs in prior studies vary from small-scale mutations (<1 kb) to large-scale mutations (1 kb–100 Mb). Targeting larger regions may come with the trade-off of increased number of false variant calls and sequencing costs and reduced number of individual cells to be sequenced [62]. Nevertheless, whole-genome sequencing (WGS) allows new discoveries of genomic variations occurring even in non-coding regions that may add significant values to the analysis of rare tumor cells.

## 3. CTC Heterogeneity and Clinical Impact

While resolving cellular heterogeneity, single-cell approaches may link specific CTC subpopulation programs to cancer cell phenotypes, metastasis, patient outcomes, and drug resistance, as demonstrated by recent studies. Examined below are genomic aberrations commonly analyzed in CTCs and their clinical impact (Figure 2). Clinical data derived from scCTC transcriptomic analyses are discussed elsewhere [14].

### 3.1. Single Nucelotide Variation (SNV)

#### 3.1.1. PIK3CA

PIK3CA is a gene harboring major driver mutations in many cancer types [63,64]. Its mutational status has increasingly been recognized as a promising predictor of resistance to targeted therapies [65]. In breast cancer, tumors harboring PIK3CA mutations are often resistant to HER2-based therapy [66,67,68], and are less likely to achieve pathologic complete response to anti-HER2 treatments [69,70]. Though limited to the analysis of EpCAM-expressing CTCs, scCTC studies have applied targeted sequencing approaches to examine mutational hotspots, most commonly in exon 9 and 20 [16,18,28,30,35,37,48,71]. The assessment of pre-existing resistant clones through scCTC analysis prior to the administration of HER2-based therapies has been suggested to be of clinical significance for patients harboring CTCs with HER2 amplification and double-mutant PIK3CA/HER2 [30]. Longitudinal monitoring of therapy response through HER2 mutational analysis of CTCs in this subset of patients will be of particular clinical interest, given the known drug efficacy of PIK3CA pathway inhibitors in patients with HER2^+^ primary tumors [72],

PIK3CA mutational status in CTCs indicative of resistance against HER2-targeted therapy has been demonstrated for HER2^−^ metastatic breast cancer patients screened for German multicentric phase III trial (i.e., DETECT III study) harboring HER2^+^ CTCs [35]. Further, studies have noted a high degree of intrapatient cellular heterogeneity and discordant PIK3CA status between CTCs and matched primary tumors [18,28,35,36], of which PIK3CA was one of two genes (among >2200 COSMIC mutations analyzed) frequently mutated in CTCs, cfDNA, and matched primary tumor in HER2^-^ breast cancer [36]. PIK3CA mutation has also been implicated in drug resistance of EGFR tyrosine kinase inhibitor (TKI) treatment. Notably, a lung cancer patient harboring PIK3CA mutation in almost all CTCs (7/8 CTCs) but not in primary tumor (low abundance) had progressive disease and presented distant metastasis after one month of treatment with erlotinib. Early detection of such resistant cells in such a less invasive way may thus be tremendously useful in drug selection.

#### 3.1.2. TP53

TP53 is a tumor suppressor gene frequently mutated in most human cancers [73]. TP53 mutations have functional implications in key molecular events in tumor progression, such as EMT [74], stemness [75], cancer prognosis, and survival outcomes [76]. Highly heterogeneous TP53 mutational status was observed across individual CTCs in prostate [24], lung [43], colorectal [16], and breast cancer [29,31,36,38,41]. In metastatic prostate cancer, ubiquitous TP53 mutations were found among multiple foci of the primary tumor and metastases, suggesting divergent cancer evolution from a single ancestor cancer [24].

In breast cancer, TP53 harbored the highest number of mutations across CTCs [31]. Mutant TP53 p.R273C has been associated with cisplatin chemotherapy resistance [77]. Concurrent mutations of RB1 and TP53 genes were also found in the majority of CTCs from a lung cancer patient who experienced a phenotypic transition from adenocarcinoma to small-cell lung cancer (SCLC) [43]. Notably, dramatic clinical response was observed in this patient upon etoposide-cisplatin treatment, which is a standard chemotherapy for SCLC patients [78,79]. These studies altogether highlight how scCTC profiling may provide an early signal of phenotypic transition in tumor to guide new therapeutic regimen.

#### 3.1.3. EGFR

Despite promising efficacy of EGFR in multiple cancer types [80], prediction of response against EGFR inhibition still remains ambiguous. The mere assessment of EGFR expression at the DNA or protein level using bulk primary tumor samples has not been an ideal indicator for predicting the response to anti-EGFR drugs [81,82]. Whole-exome sequencing of lung CTCs revealed specific INDEL in the EGFR gene (p.Lys746_Ala750del) shared by primary tumors and metastases, which could be targeted with TKIs [43]. In our earlier work, we reported highly sensitive detection of EGFR mutations (T790M and L868R) in microfluidically enriched CTCs, which showed a complete concordance of mutation status with matched primary tumors in non-small-cell lung cancer (NSCLC) patients [44]. Similarly, an integrated method using a magnetic sifter (MagSifter) and nanowell system has been described for accurate detection of EGFR (del19, T790M, and L868R) mutations in CTCs from lung cancer patients [22]. Notably, RT-qPCR readings of bulk blood samples did not reach detectable level to be analyzed for same mutations in this study [22].

#### 3.1.4. KRAS

KRAS, one of the genes involved in the EGFR signaling pathway, may have predictive value of clinical response to anti-EGFR therapies, such as cetuximab [83], panitumumab [84], and gefitinib [85]. Patients exhibited mutational disparity between CTCs and matched primary tumors and/or across individual CTCs in breast [36], colorectal [47,48,49], and multiple myeloma [24], and breast cancer [36]. This may explain the variable response to anti-EGFR treatment in these cancer patients. Similarly, a highly varying degree of concordance in KRAS mutational status between CTCs and primary/metastatic lesions was observed, reflecting intratumoral heterogeneity of point mutations in KRAS occurring in 48–76% of various cancers [86].

#### 3.1.5. BRAF

Another mutation predictive of response to EGFR-inhibiting therapy is BRAF, which is associated with a very poor prognosis particularly in colorectal cancer and melanoma [87]. Further, the predictive values of V600E and V601E mutations have been demonstrated for the use of RAF kinase inhibitor (vemurafenib) and MEK inhibitor (trametinib), respectively, in BRAF-mutated melanomas [50,88]. Somatic missense mutations in BRAF have been found in approximately 10% and 60% of colorectal tumors and melanoma lesions, respectively [50,89]. Single-cell genomic characterization of CTCs across these cancer types have revealed highly heterogeneous BRAF status across CTCs [47,48,50,51], with disparities in BRAF mutations to the corresponding primary tumor [47,50]. While such considerable heterogeneity observed in these selected genes might be the result of newly acquired mutations in CTCs, it is likely that these mutations were missed by single sector-based tissue biopsies. Through the additional deep sequencing of tissue samples, other groups indeed found mutations that were initially unique to CTCs in the primary tumors and metastases at subclonal level [16].

### 3.2. Microsatellite Instability (MSI)

Defective DNA mismatch repair (MMR) machinery leads to hypermutation and instability in nucleotide repeat sequences [90]. MSI is an established prognostic [91,92,93] and predictive marker [94,95] in many cancer types. In colorectal cancer (CRC), tumors with high-level MSI, or MSI-H phenotype account for ~15% of metastatic disease [96], and have distinct pathologic and clinical features [97]. MSI typing may serve as a predictor of benefit from adjuvant 5-fluorouacil chemotherapy for non-MSI-H CRC patients [95,98]. The standard MSI assessment recommended by the National Cancer Institute/International Collaborative Group/HNPCC (NCI/ICG-HNPCC) involves the examination of two mononucleotide repeats and three dinucleotide repeats in tumor and non-tumor adjacent normal tissues [99].

A comprehensive genomic study of CRC identified microsatellite stable (MSS) tumor harboring MSI CTCs, through aCGH, mutational profiling, and MSI analyses at the single-cell level [47]. Of note, immunohistochemical (IHC) expression analysis of DNA MMR proteins (i.e., MLH1, MSH2, MSH6, and PMS2) in multiple sectors obtained from matched primary tumor and metastatic lesions did not indicate MSI-H status in this patient. Such discordance was also found in mutational profiles, where mutations in key genes such as KRAS and TP53 were detectable only in CTCs but not in the tumor. Given that contaminating stromal cells or surrounding non-tumor cells in the tissue may obscure MSI, single-cell approaches will be essential for MSI typing.

### 3.3. Copy-Number Variation (CNV)

With the advent of SNP/aCGH arrays and NGS technologies, identification and characterization of CNVs in primary and metastatic tumors have provided insight into the role of CNVs in cellular functions and cancer pathogenesis [100,101,102]. Gain of oncogenes and loss of tumor suppressors are frequent drivers of tumor progression, and are closely associated with therapeutic responses [103]. In CTCs, genome-wide CNVs were found to be highly reproducible from cell to cell within the same individual and across different lung cancer patients with same pathological subtypes [43,45]. This is consistent with homogeneity found in CTCs from SCLC [42,46] and colorectal cancer patients [45].

While harboring a substantial number of genomic aberrations, CTCs exhibit concordant changes in chromosomes observed in matched primary and/or metastatic tumors, supporting their malignant origin, but to different extents across individual CTC, as demonstrated in colorectal [45,47], breast [40], and bladder [52] cancers. For example, phylogenetic analysis of CNV profiles identified homozygous deletion of a chromosomal region containing the tumor suppressor gene, PTEN, in CTCs and lymph node metastases, but not in primary tumor, suggesting that such CTC subsets might have metastasized in colorectal cancer [45].

A unique signature of recurrent CNVs specific to CTCs was also found in breast cancer, consisting of genes and miRNAs related to CTC phenotypes, such as resistance to anoikis, TGFβ signaling, and metastasis [23]. This copy number signature clustered patients into two groups, independent of subtype, revealing distinct functional or metastatic features in different populations. Gain of a chromosomal region harboring the HER2 gene was further consistently observed across CTCs regardless of HER2 status of matched primary tumors in this study, suggesting a potential role of HER2 amplification in CTC biology. Notably, HER2 amplification associated with the degree of chromosomal changes was identified in another multi-scale scCTC study, where a significantly higher number of genomic rearrangements was observed in breast CTCs with HER2 amplification than those without amplification, whereas no such change was seen for PIK3CA mutations [30].

CNV profiling of CTCs could potentially be used as a tool for risk stratification. CNV-based classifiers that can assign SCLC patients as chemosensitive or chemorefractory have been developed [42,46], one of which have been validated in an independent patient cohort [42]. The two patient groups stratified by these classifiers were found to have significant different progression-free survival (PFS) and/or overall survival (OS), demonstrating prognostic and predictive value of CTC CNV profiles [42,46]. Further, apheresis-acquired, CK^+^ CTCs harboring >10 chromosomal alterations were associated with risk of early metastasis [32]. Better therapeutic strategies may also be inferred from sequential single-cell characterization of CNV changes in CTCs over the course of treatment; MYC amplification occurred along with AR protein expression and AR amplification in a prostate cancer patient progressing through targeted therapy [27]. While direct targeting of MYC proves to be difficult, scCTC data suggest that co-targeting of c-Myc in conjunction with AR may serve as an alternative path to prevent the emergence of drug-resistant subclones.

### 3.4. Chromosomal Breakpoints

In addition to focal CNV analyses, whole genome-wide CNV profiles have been extensively analyzed in bulk primary tumors. They are often associated with therapeutic resistance against platinum-based chemotherapy and PARP inhibitors [104]. Large-scale state transition (LST), which is defined as the number of chromosomal breaks between adjacent regions of at least 10Mb, is one of surrogates of such large-scale genomic instability. Significantly higher and heterogeneous LST scores were observed in CTCs from metastatic castration resistant prostate cancer (mCRPC) patients compared to cancer cell lines and WBCs from healthy donors, implying unstable CTC genomes at the single-cell level [25]. The ability to assess genomic instability with LST scoring could potentially improve risk stratification for therapeutic strategies.

### 3.5. Chromosomal Rearrangement

Two SVs common in all CTCs and both primary and metastatic tumors were found in prostate cancer [26]: TMEM207 in chr3, which facilitates tumor invasion, migration, and metastasis [105,106], and chr13-chr15 translocation. An extensive assessment further revealed heterogeneous status of SVs involving tumor suppressor genes, such as BRCA2, RB1, and PTEN, which are common in prostate cancer [107]. Heterogeneous status of PTEN SVs in single CTCs may suggest the acquirement of variations at different time-points, as PTEN point mutations emerges as a late event during cancer evolution [108]. Parallel transcriptomic profiling for detecting such SVs encoding oncogenes or tumor suppressors that carry founder mutation will further allow discovery of novel fusion products.

## 4. Longitudinal Studies

While it has long been recognized that CTC count could serve as a robust predictor of patient outcomes in breast, prostate, and colorectal cancers [36,39,109,110,111,112], CTC enumeration at a single time point alone may not provide sufficient information in terms of treatment regimen. Alternatively, phenotyping of CTCs with morphometric parameters and protein expression has been shown to be better correlated with therapy response [27,113]. Another major advantage of CTC profiling is the feasibility to analyze longitudinal samples to study mechanisms related to acquired resistance to therapy (Figure 3). Yet, such analyses may be challenging in the absence of evident de novo global chromosomal changes upon relapse and progression, requiring larger patient cohorts [42].

Selection of patients for targeted therapy has been based on the IHC detection of protein of interest and/or fluorescence in situ hybridization-based analysis of gene-level alterations, typically in known oncogenes, using primary tumor tissues. An underlying assumption is that only marker-positive patients will respond to mutant oncogene-targeted therapy. These treatments may include vemurafenib (BRAF inhibitor) for melanoma patients with V600E/V600K mutation in the BRAF gene [114], Neratinib (HER2 TKI) [115], trastuzumab (HER2 antibody) [116] and pertuzumab (HER2 antibody) [117] for breast cancer patients with amplification and/or overexpression of the HER2 gene, and erlotinib (EGFR TKI) for non-small cell lung cancer patients with exon 19 deletion or L858R mutation in the EGFR gene [118].

Such oncogene-targeted therapies, however, should carefully be applied in the clinical settings. IHC detection of EGFR expression in bulk tumor tissue alone, for example, may not be an ideal tool for prediction of response to gefitinib [119]. Discordance in genomic profiles between primary and recurring/metastatic tumors [120,121,122] attributed to clonal changes, sampling error in clonal selection, and/or technical flaws in the assay has complicated the decision-making process in treatment selection. Repeat biopsy for marker reassessment does not guarantee improved accuracy, nor is it without false-negative readings [123].

Varying genomic status of oncogenes has thus been re-evaluated with CTC pools over the course of therapy. Studies have examined the feasibility of detection of HER2^+^ CTCs in patients with HER2^–^ primary tumors [124,125] and that of KRAS-mutated CTCs in patients with nonmutated colorectal primary tumor and mutated metastases [89]. Notably, it was found that treatment with trastuzumab, an anti-HER2 antibody traditionally prescribed for patients with HER2^+^ primary tumor, was shown to be effective in improving survival outcomes of patients with HER2^−^ tumor by eliminating CK^+^HER2^+^ CTCs [126]. This was the first study that clearly demonstrated the potential of CTCs to effectively monitor evolving mutational landscape reflecting time-varying changes in drug susceptibility. Since then, several prospectively conducted studies have been carried out to facilitate the phenotyping and genetic characterization of CTCs as targeted-therapeutic intervention in metastatic cancer [127].

Longitudinal genomic data on SNVs and CNVs have so far been obtained in scCTC studies where CTCs were collected in a sequential manner during the course of treatment in lung [42,43] and prostate cancer [17]. A unique set of somatic CNV variations distinct from those observed in response to standard chemotherapy, was identified in single prostate CTCs at the time of targeted therapy failure, inferring rapidly evolving genomic organization and the emergent putative-resistant clones [27]. In another study where single-cell characterization was done before and during treatment for a breast cancer patient, while CTCs revealed distinct mutational profiles at different time points, all CTCs harbored mutation in the HER2 gene (p.V777L), regardless of sampling time, indicating resistance to HER2-targeted treatment in this patient. Interestingly, the best treatment response to chemotherapy with capecitabine and vinerolbine was observed in a patient who had the highest number of mutated genes and sequence variants in single breast CTCs [31].

Not all longitudinal studies, however, have demonstrated a clear association between CTCs and metastatic status. PIK3CA mutational status in majority of breast CTCs (7 out of 9 blood draws) sequentially sampled over time was not reflective of bone and lung metastases while DTCs achieved 100% concordance in a patient with progressive metastatic breast cancer [28]. Such highly discordant results, however, may be attributed to EpCAM-based approach employed in this study for CTC enrichment, missing out de-differentiated EpCAM^−^ or mesenchymally shifted CTCs, all of which may be major constituents of putative metastatic founders [4].

## 5. Challenges and Emerging Technologies

Clearly, recent CTC studies have uncovered new perspectives in tumor biology and cancer management, through the application of rare cell sorting and single-cell sequencing technologies. Practically, however, there are challenges and technical errors associated with the developed workflows presented in this work.

### 5.1. Fresh-Frozen Versus Formalin-Fixed

Immunoaffinity-based enrichment technologies, including CellSearch^®^, involve a fixation step, which makes use of fixatives to stabilize whole blood for up to 96 h. Yet, fixed CTCs may be not suitable for RNA-based measurements, ex vivo culture and expansion, drug screening, and xenograft model-based functional studies, for compromised cell viability and degraded RNAs. Alternatively, multiparametric flow cytometry or FACS can be used to keep viable CTCs, which can further be expanded ex vivo to generate patient-derived 3D-spheroids, as demonstrated in prostate and breast cancers [17,34]. DEPArray^TM^ and microfluidic technologies can also be used to isolate viable CTCs for subsequent molecular screening on a cell-per-cell basis.

To preserve cell viability, peripheral blood sample should be delivered on ice immediately to the laboratory once collected from a cancer patient. RNA degradation occurs within 2–4 h, and sample processing >5 h after the blood draw may result in >60% loss in CTC yield [128]. Similar to tissue acquisition of solid lesions, these requirements impose practical challenges in hospitals and labs, particularly for longitudinal cohort studies. Nevertheless, emerging technologies may be applied to CTCs to circumvent such issues, through the use of nuclear RNA or preservation protocol that retains cell viability and RNA quality for up to 72 h for single-cell transcriptomic profiling. Studies have demonstrated high concordance between nuclear RNA and whole cell RNA in the expression of cell-type specific and metabolic modeling genes [129], and that between fresh and preserved blood in detecting cancer-specific transcripts [128]. Alternatively, frozen-optimized scRNA-seq protocols (e.g., Nuc-seq) may be applied to CTC profiling, which will be particularly useful for serial monitoring of previously inaccessible tissues [2].

### 5.2. Increasing Number of CTC Libraries

Among all the steps, WGA was found to be the most error-prone step in performing scCTC sequencing, with allele dropout (ADO) being a significant source of failure for scCTC sequencing [53]. Such prevalent limitations may be overcome if the amount of DNA template is increased. For example, it has been suggested that at least 10 CTCs are required to reliably detect point mutations in KRAS from pancreatic CTCs [53]. Similarly, sequencing multiple independent libraries of CTCs has been proposed to improve sensitivity in determining variants and to better represent bulk library from a matched primary tumor in prostate cancer [24]. The development of WGA methods providing improved uniformity in genome-wide coverage of the amplified DNA may further facilitate reproducible and accurate sequencing for clinical use.

Alternatively, highly sensitive CTC enrichment technologies may be applied to scCTC sequencing studies to capture, assumedly, all CTCs present in blood so as to generate more DNA templates. Recently, methods of performing single-cell CNV analysis of CTCs acquired by apheresis was described in prostate and breast cancers [17], where CTCs harvested from an apheresis (mean volume = 59.5 mL) achieved approximately 90-fold increased yield (CTC count = 12,546) [17]. Extrapolation analysis further indicated that CTCs might be concentrated along with the mononuclear cell populations during diagnostic leukapheresis (DLA), thereby enhancing CTC detection frequency even in nonmetastatic cancer patients, as compared to standard CTC blood tests processing a volume of 1–10 mL of peripheral blood [32].

In recent years, scRNA-seq technologies coupled with massively parallel microfluidics have enabled high-throughput analysis of mouse retinal cells [130], human macrophages [131], embryonic stem cells [132], and peripheral blood mononuclear cells (PBMCs) [133]. Their application to CTCs, however, has been relatively limited due to the inefficiency of bead-cell pairing and the inevitable contamination of blood cells even in primarily enriched samples [134]. To overcome these limitations, Cheng et al. have recently developed Hydro-seq, which enable high-throughput contamination-free scRNA-seq of CTCs from breast cancer patients, uncovering cellular heterogeneity in metastasis and therapy related genes [134]. Having the scale-up capability while achieving high cell-capture efficiency and high-fidelity single-cell sequencing results, these technologies will play an increasingly important role in future studies aiming to generate more accurate and reproducible data at high throughput with low cost.

### 5.3. Multidimensional Measurements

While others have attributed the appearance of CK^+^CD45^+^ cells to false-positive CK^+^ staining of WBCs [32], such “double-positive” cells may have functional roles or clinical implication within circulation given their occurrence at a much lower frequency in healthy blood samples [135]. Similarly, the exact role of apoptotic CTCs (e.g., CK^+^CD45^−^ cells with abnormal chromosomal patterns and/or nuclear fragmentation), PD-L1^+^CD45^−^ cells, and CK^–^CD45^−^ cells in metastasis is not yet fully clear. Technologies enabling joint profiling of multiple modalities from the same individual cell may provide accurate means to understand clinical implication on the occurrence of these specific CTC subpopulations in circulation.

The high definition-CTC (HD-CTC) is an exemplary technology that facilitates real-time single-cell characterization of morphometric (i.e., cell roundness, cell area, AR subcellular localization) and protein expression changes in AR for prostate cancer [27]. Similarly, the functional EPISPOT assay, which is now named EPIDROP, allows simultaneous single-cell analysis of proteome and secretome of viable CTCs or of CTC clusters [136]. Such functional assays or microfluidic technologies, which have been successfully applied to patient-derived CTCs for analysis of genome [44], transcriptome [4], proteome [137], metabolome [138], and secretome [139] at the single-cell level, may further be integrated into the framework for deriving multidimensional data. Along with the increasingly available genomic data derived from tissue biopsies spanning diverse cancer types [140,141], the combined analysis of tissue and liquid biopsies may further uncover new insights into tumor heterogeneity and provide additional clinical information, as recently shown in lung cancer [4].

## 6. Conclusions

Recent scCTC studies have focused on how best to (1) identify and isolate extremely heterogeneous, fragile, and rare CTCs in a highly specific and unbiased manner, (2) discriminate false negatives and detect actionable mutations, and (3) relate the findings to clinically meaningful outcomes that could not have been caught by a tissue biopsy. Emerging data pointing towards the prevalence of CTC subpopulations and their differing metastatic potential have further stimulated studies aiming to identify and genetically/phenotypically characterize such premetastatic subsets of CTC populations that are favored to be liberated from primary tumors, survive in the bloodstream, and succeed in the early colonization phases.

While past efforts in deconvoluting the complex nature of CTCs have been largely ineffective with bulk-cell analysis, single-cell approaches are beginning to unmask cellular heterogeneity of CTCs and their clinical significance, providing a foundation for liquid biopsy in the clinic. As we continue to develop sensitive CTC enrichment technologies and generate more sequencing data from patient-derived CTCs, clinicians will continue to find a better way to apply liquid biopsies, possibly (1) at initial diagnosis, for prognostication, (2) after tumor resection, for assessment of residual disease, and (3) after adjuvant therapies, for prediction of early recurrence or relapse. With mounting molecular evidence suggesting prognostic value of CTC-derived biomarkers predictive of response to chemotherapy, targeted therapy, and immune checkpoint inhibitors, we anticipate that, in the near future, liquid biopsies will become a routine screening and monitoring of cancer patients.

## Figures and Tables

**Figure 1 cancers-11-01595-f001:**
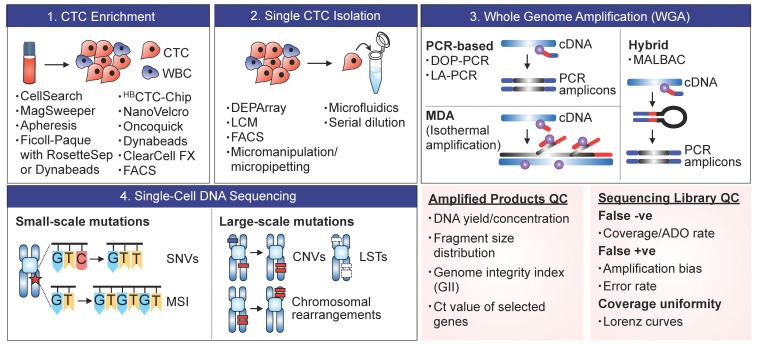
The standard workflow and existing technologies for scCTC sequencing. CTCs are primarily enriched from a whole blood sample, and are isolated as single cells for subsequent downstream molecular analyses. For genomic analysis of whole genome/exome, whole-genome amplification (WGA) is performed and amplified DNA products are QC-checked prior to sequencing.

**Figure 2 cancers-11-01595-f002:**
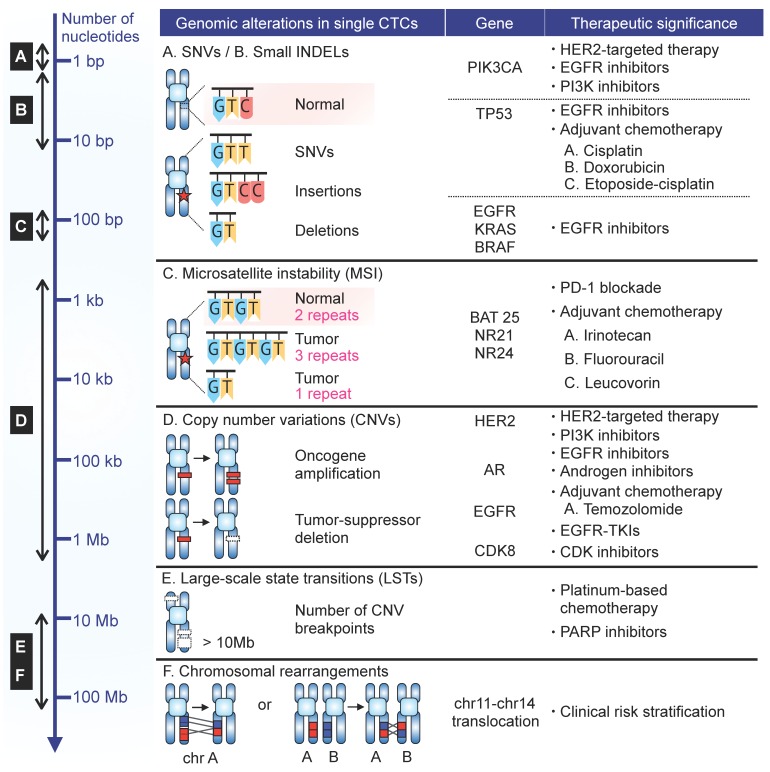
Summary of genomic alterations found in scCTC sequencing studies.

**Figure 3 cancers-11-01595-f003:**
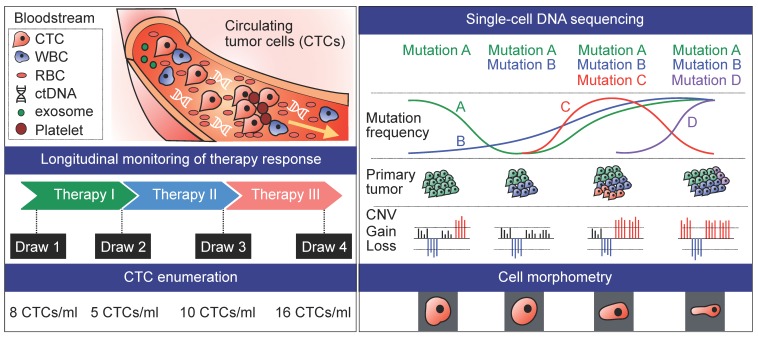
scCTC DNA sequencing for longitudinal monitoring of therapy response.

**Table 1 cancers-11-01595-t001:** Summary of single-cell circulating tumor cells (scCTC) sequencing studies that analyzed DNA alterations in patient-derived CTCs.

CTC Enrichment	Single-Cell Isolation	CTC Criteria	WGA	Profiling	Investigated Genes	Genomic Analysis	Number of Single CTCs (Patients) ^1^	Ref.
Prostate cancer								
MagSweeper	CellCelector	DAPI− CD45− EpCAM+	MDA	NGS	All	SNVs	42 (5)	[24]
Epic Sciences CTC Platform	Micromanipulation	CD45− CK+/−	DOP-PCR	NGS	All	CNVs, LSTs	67 (7)	[25]
NanoVelcro CTC Chip	LCM	CD45− CK+	MDA	NGS, Sanger, aCGH	All	SNVs, SVs, CNVs	12 (1)	[26]
HD-CTC Assay	Micromanipulation	DAPI+ CK+ CD45−	LA-PCR	NGS	All	CNVs	41 (1)	[27]
CellSearch, Spectra Optia Apheresis System	FACS	DAPI+ CK+ CD45−	LA-PCR	aCGH	All	CNVs	205 (14)	[17]
Breast cancer								
MagSweeper	Micromanipulation	DAPI+ CK+ CD45−	No WGA	Sanger	PIK3CA	SNVs	185 (11)	[28]
CellSearch	DEPArray	DAPI+ CK+ CD45−	LA-PCR	Sanger	TP53	SNVs	11 (2)	[29]
CellSearch	DEPArray	DAPI+ CK+ CD45−	LA-PCR	Sanger	PIK3CA	SNVs	241 (43)	[30]
				qPCR	HER2	CNVs	192 (42)	
				aCGH	All	CNVs	37 (15)	
CellSearch	DEPArray	DAPI+ CK+ CD45− CD34−	LA-PCR	Sanger	PIK3CA	SNVs	115 (18)	[18]
CellSearch	DEPArray	DAPI+ CK+ CD45−	LA-PCR	Targeted NGS	50 cancer-related genes	SNVs	14 (4)	[31]
Leukapheresis, CellSearch	Micromanipulation	CK+ CD45−	DOP-PCR	CGH	All	CNVs	65 (19)	[32]
CellSearch	MoFlo XDP flow-sorting	DAPI+ CK+ CD45−	LA-PCR	aCGH	All	CNVs	26 (12)	[33]
				qPCR	CCND1 locus	CNVs		
				Sanger	PIK3CA, TP53	SNVs		
FACS	DEPArray	DAPI− CD45− EpCAM− CD44+ CD24− uPAR+/− intβ1+/−	LA-PCR	MassARRAY	>200 hallmark cancer genes	SNVs	7 (-)	[34]
CellSearch	Micromanipulation	DAPI+ CK+ CD45−	MDA, LA-PCR	Sanger	PIK3CA	SNVs	114 (33)	[35]
CellSearch	DEPArray	DAPI+ CK+ CD45−	LA-PCR	Targeted NGS, ddPCR	50 COSMIC genes	SNVs	40 (5)	[36]
CellSearch	CellCelector	DAPI+ CK+ CD45−	LA-PCR	Targeted NGS	50 COSMIC genes	SNVs	7 (2)	[37]
CellSearch	DEPArray	DAPI+ CK+ CD45−	LA-PCR	Sanger	TP53, HER2, PIK3CA, RB1	SNVs	24 (6)	[38]
Ficoll Separation	Micromanipulation	DAPI+ CK+ CD45−	LA-PCR	Sanger	ESR1	SNVs	8 (4)	[39]
AutoMACS Classic Separator	LCM	CK+ CD45−	LA-PCR	SNP Array	All	CNVs	17 (17)	[23]
CellSearch, Oncoquick	CellCelector	EpCAM+ CD45−	MDA	aCGH, targeted NGS	All	SNVs, CNVs	31 (1)	[40]
oHSV-hTERT-GFP method	FACS	CD45− hTERT+	MALBAC	NGS	All	SNVs, CNVs	11 (8)	[19]
ScreenCell	DEPArray	DAPI+ CK+ CD45−	LA-PCR	Sanger	TP53, ESR1	SNVs	7 (1)	[41]
ClearCell FX System	Manipulation	DAPI+ CK+ CD45−	MALBAC	NGS	All	SNVs	3 (1)	[20]
Lung cancer								
CellSearch	DEPArray	DAPI+ CK+ CD45−	LA-PCR	NGS	All	CNVs	72 (13)	[42]
CellSearch	Micromanipulation	DAPI+ CK+ CD45−	MALBAC	NGS, digital PCR, Sanger	All	SNVs, INDELs	24 (4)	[43]
						CNVs	61 (11)	
ClearCell FX System	Microfluidic chip	DAPI+ CK+ CD45−	No WGA	Sanger	EGFR	SNVs	26 (7)	[44]
CellSearch	Micromanipulation	DAPI+ CK+ CD45−	MALBAC	NGS	All	SNVs, INDELs, CNVs, SVs	97 (23)	[45]
CellSearch	Micromanipulation	DAPI+ CK+ CD45−	MALBAC	NGS	All	SNVs, INDELs, CNVs, SVs	91 (10)	[46]
MagSifter	Single-Nanowell Assay	DAPI+ CK+ CD45− TERT+ MET+	No WGA	Multiplex PCR	EGFR	SNVs	202 (7)	[22]
Colorectal cancer								
CellSearch	Micromanipulation	EpCAM+ CD45− CK+	LA-PCR	aCGH	All	CNVs	8 (8)	[47]
				Sanger	KRAS, BRAF, TP53	SNVs	126 (31)	
				Multiplex PCR	NCI/ICG-HNPCC marker panel	MSI	122 (30)	
CellSearch	Micromanipulation	DAPI+ CK+ CD45−	LA-PCR, MDA	aCGH	All	CNVs	37 (6)	[16]
				Targeted NGS	68 colorectal cancer-associated genes	SNVs	8 (2)	
CellSearch	Micromanipulation	DAPI+ CK+ CD45−	LA-PCR, MDA	qPCR	EGFR	CNVs	26 (3)	[48]
				Sanger	KRAS, BRAF, PIK3CA	SNVs	69 (5)	
Oncoquick	DEPArray	HOECHST+ CK+ CD45−	LA-PCR	Sanger, pyrosequencing	KRAS	SNVs	- (16)	[49]
Melanoma								
Dynabeads	LCM	HMW− MAA+ CD45− MART-1/gp100+	No WGA	Sanger	BRAF	SNVs	14 (9)	[50]
					KIT	SNVs	4 (4)	
Dielectrophoretic microwell array	Micromanipulation	CD45− MART-1/gp100+	No WGA	Sanger	BRAF	SNVs	33 (1)	[51]
Multiple myeloma								
MACS beads	Micromanipulation	CD45−CD138+	MDA	Targeted NGS	35 most commonly mutated loci	SNVs	203 (10)	[21]
Epic Sciences CTC Platform	Micromanipulation	CK+/− CD45−	DOP-PCR	NGS	All	CNVs	9 (1)	[52]
Pancreatic cancer								
NanoVelcro Chip	LCM	HOECHST+ CD45− CK/CEA+	MDA	Sanger	KRAS	SNVs	60 (12)	[53]

^1^ QC-passed CTCs that have been included in the final analysis.

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
