# Peer review of "Single-Cell Analysis of Circulating Tumor Cells: Why Heterogeneity Matters"

_cancers, 2019, doi:10.3390/cancers11101595_

Round 1

Reviewer 1 Report

Lim et al.wrote a review on single-cell analysis of circulating tumor cells (CTCs) and why heterogeneity matters. The authors reviewed nicely recent single-cell CTC sequencing efforts, and the integrated workflows, that have successfully characterized patient-derived CTCs.

The article is well written and has different chapters: (1) Introduction; (2) Methods and technologies with a Table summarizing single CTC sequencing studies (in breast, prostate, lung, colorectal, melanoma, multiple myeloma, pancreatic cancers) that analyzed patient-derived CTCs; (3) CTC Heterogeneity and Clinical Impact; (4) longitudinal studies; (5) Challenges and Emerging Technologies plus (6) Conclusion.

Different points to be addressed:

Page 2 – line 47: This paragraph should be re-written. “High degree of ITH observed within the tumor is likely to be reflected in circulating tumor cells (CTCs), which are derived from either “benign” cells that can passively enter the bloodstream or actively invading cancer cells that have been mesenchymally-shifted in primary or metastatic tumor through epithelial-mesenchymal transition (EMT).”

First, the notion of release of CTCs by benign disease should be written more clearly and with precaution as it has been observed only in special and rare cases. A reference should be given to support this statement. In addition, the EMT is not required for all tumor cells to be invasive, thus it should be mentioned differently, also with a reference to support this statement.

Page 5 – line 85: in CTC enrichment, the leukocyte depletion using the RosetteSep technology is missing.

Page 12: in the chapter “Challenges and Emerging Technologies”, the EPISPOT assay and the new optimized EPIDROP assay could be mentioned as a new emerging technology (ref Pantel & Alix-Panabières, Nat Rev Clin Oncol 2019). It is a novel functional assay allowing the detection of viable CTCs at the single cell level.

Reviewer 2 Report

The review entitled "Single-cell analysis of circulating tumor cells: Why heterogeneity matters" is very interested, well written, documented and illustrated. It deserves to be published but I have 2 minor points:

In the single nucleotide variation list BRAF mutation needs to be added since it is a very common mutation with a very bad prognosis in various cancers such as colorectal cancer. A very important paper in the field had just come out on breast cancer and has to appear in this review (Cheng YH, Nature Communications 2019)
